# Reconstitution defines the roles of p62, NBR1 and TAX1BP1 in ubiquitin condensate formation and autophagy initiation

Eleonora Turco [1,2✉], Adriana Savova[1,2], Flora Gere[1], Luca Ferrari[1], Julia Romanov[1], Martina Schuschnig[1] & Sascha Martens [1✉]

The autophagic degradation of misfolded and ubiquitinated proteins is important for cellular homeostasis. In this process, which is governed by cargo receptors, ubiquitinated proteins are condensed into larger structures and subsequently become targets for the autophagy machinery. Here we employ in vitro reconstitution and cell biology to define the roles of the human cargo receptors p62/SQSTM1, NBR1 and TAX1BP1 in the selective autophagy of ubiquitinated substrates. We show that p62 is the major driver of ubiquitin condensate formation. NBR1 promotes condensate formation by equipping the p62-NBR1 hetero-oligomeric complex with a high-affinity UBA domain. Additionally, NBR1 recruits TAX1BP1 to the ubiquitin condensates formed by p62. While all three receptors interact with FIP200, TAX1BP1 is the main driver of FIP200 recruitment and thus the autophagic degradation of p62–ubiquitin condensates. In summary, our study defines the roles of all three receptors in the selective autophagy of ubiquitin condensates.

[1] Max Perutz Labs, University of Vienna, Vienna BioCenter (VBC), Vienna, Austria. [2]These authors contributed equally: Eleonora Turco, Adriana Savova. ✉email: eleonora.turco@univie.ac.at; sascha.martens@univie.ac.at

Macroautophagy (hereafter autophagy) is a conserved intracellular process, which aids cellular homeostasis by the disposal of harmful structures including protein aggregates, damaged organelles, and intracellular pathogens[1–3]. Defects in autophagy have been linked to a plethora of diseases including cancer and neurodegeneration[4]. The degradation of the harmful material, referred to as cargo, is mediated by its encapsulation within de novo formed double-membrane vesicles, termed autophagosomes, which fuse with lysosomes wherein the cargo is degraded. Autophagosome formation is mediated by the autophagy machinery[5,6]. In selective autophagy, during which specific cargo is targeted for elimination, this machinery is recruited by cargo receptors such as p62/SQSTM1, NBR1, NDP52, and optineurin[7–13]. Many of the various cargo receptors in mammalian cells, including p62 and NBR1, recognize the cargo via its ubiquitin tags[14].

A major function of p62 is the degradation of ubiquitinated, misfolded proteins by selective autophagy. In this process, it plays several roles[15–17]. First, it mediates the condensation of ubiquitinated proteins into larger structures[18–20]. Subsequently, it contributes to the local formation of autophagosomes around these condensates by recruiting the FIP200 scaffold protein[10]. Finally, it links the cargo to the nascent autophagosomal membrane via its interaction with LC3 and GABARAP proteins, which decorate the forming autophagosomal membrane[21,22]. p62 oligomerizes into filaments through its N-terminal PB1 domain[23–25]. This oligomerization is required for its ability to form condensates with ubiquitinated proteins but also to avidly bind the LC3/GABARAP decorated autophagosomal membrane via its LC3 interacting region (LIR) motif and the ubiquitinated cargo via its C-terminal UBA domain (Supplementary Fig. 1a)[19,20,26].

In these processes, p62 is aided by the cargo receptor NBR1. Similar to p62, NBR1 binds LC3/GABARAP proteins via LIR motifs and ubiquitin via its UBA domain[27]. NBR1 and p62 directly interact through their N-terminal PB1 domains (Supplementary Fig. 1a)[20,24,25]. In cells, NBR1 colocalizes with p62, and its depletion results in fewer p62 condensates[27,28]. In vitro, NBR1 directly enhances the formation of p62–ubiquitin condensates[20]. The specific mechanisms through which NBR1 promotes condensate formation and thus cargo degradation remain unclear.

A third cargo receptor, TAX1BP1 (Supplementary Fig. 1a), was recently shown to colocalize with NBR1[13]. TAX1BP1 is required for the clearance of protein aggregates[29]. It interacts with NBR1 to recruit FIP200 and to trigger its subsequent degradation in LC3 lipidation independent autophagy[13].

Here we employ in vitro reconstitutions as well as cell biological approaches to show that p62, NBR1, and TAX1BP1 cooperate during the formation and degradation of p62–ubiquitin condensates by selective autophagy. We found that p62 is the major driver of ubiquitin condensate formation. We further show that NBR1 promotes p62–ubiquitin condensate formation via its PB1 domain-mediated binding to p62. This interaction equips the p62—NBR1 heterooligomeric complex with a high-affinity UBA domain provided by NBR1, allowing for more efficient cargo recognition. In addition, NBR1 serves to recruit TAX1BP1 to the condensates. While all three cargo receptors are able to interact with the FIP200 scaffold protein to initiate autophagosome formation, TAX1BP1 is the main driver for its recruitment to the condensates. In this study we disentangle the individual contributions of the three mammalian cargo receptors p62, NBR1, and TAX1BP1 in the clearance of ubiquitin condensates through selective autophagy.

## Results

### Colocalization of p62, NBR1, and TAX1BP1 in ubiquitin-containing condensates.
To determine the mechanisms of action of the p62, NBR1, and TAX1BP1 cargo receptors and their cooperation in the formation and degradation of ubiquitin-containing condensates, we studied their colocalization in HAP1 cells expressing endogenously tagged GFP-p62 and mScarlet-AID-NBR1 (Fig. 1a and Supplementary Fig. 1 b–d). In addition to a fluorophore, NBR1 was also tagged with an auxin-inducible degradation (AID) tag to allow its acute depletion on a protein level (Supplementary Fig. 1b)[30]. The tags did not interfere with the lysosomal delivery of p62 and NBR1, as their levels were increased upon inhibition of lysosomal activity with bafilomycin treatment (Fig. 1b and Supplementary Fig. 1d)[20]. Even in resting HAP1 cells, i.e., in the absence of known activators of autophagy such as starvation, treatment with drugs or overexpression of aggregation-prone proteins, p62 forms multiple dynamic condensates (Supplementary video 1). When we stained the HAP1 GFP-p62, mSc-AID-NBR1 cells with an anti-TAX1BP1 antibody (Fig. 1a), we noticed that about 13% of TAX1BP1 puncta colocalized with p62 and NBR1 (Fig. 1b). A similar percentage of p62 puncta colocalized with NBR1 and TAX1BP1 (Fig. 1c). Both these numbers increased upon bafilomycin treatment suggesting that the receptors are degraded in lysosomes (Fig. 1b, c). Consistent with previous findings[29], TAX1BP1 puncta colocalized with ubiquitin (Supplementary Fig. 1e) and with LC3B (Fig. 1d).

### NBR1 modulates p62–ubiquitin condensate formation in vitro.
Next, we went on to dissect the roles of the three cargo receptors in the formation and clearance of ubiquitin condensates. Since the depletion of p62 results in a severely reduced number or even complete absence of ubiquitin condensates[18,29], its functional interaction with NBR1 and TAX1BP1 in this process is difficult to study in cells. Therefore, we turned to a reconstituted system[10,20]. Consistent with our previous results, we observed that NBR1 alone did not induce condensation (Supplementary Fig. 2a) but that it markedly enhanced the condensation of p62 and GST fused to four M1-linked ubiquitin moieties (GST-4xUb), as judged from the number (left panel) and size (right panel) of the condensates (Fig. 2a–c and Supplementary Fig. 2a, b)[20]. NBR1 was efficiently recruited to p62-containing condensates (Fig. 2b). In order to test if there was an optimal molar ratio of NBR1 to p62 with respect to its condensation promoting activity, we titrated NBR1 into condensate formation assays containing p62 and GST-4xUb (Fig. 2d and Supplementary Fig. 2c). We observed that the promoting activity of NBR1 gradually increased with increasing concentrations, but that it dropped at an equimolar ratio. This suggested that a sub-stoichiometric concentration of NBR1 relative to p62 is ideal for the promotion of condensate formation.

Next, we asked if NBR1 would also promote the condensation of in vitro synthesized K48- and K63-linked ubiquitin chains (Fig. 2e and Supplementary Fig. 2d). NBR1 promoted the formation of condensates of these chain types as well. There are differences in the degree to which the condensation of the three substrates was promoted, but due to the different length of the chains (Supplementary Fig. 2d), we cannot distinguish between chain type and chain length. We conclude that NBR1 can enhance the condensation of a broad spectrum of substrates by p62, which is consistent with a low degree of ubiquitin linkage specificity of its UBA domain[31].

We went on to dissect which properties of NBR1 are required to promote condensation. To this end, we expressed and purified two deletion mutants of NBR1 (Supplementary Fig. 2e). The first mutant lacked the PB1 domain (NBR1ΔPB1) and, as expected from previous studies[24,25], showed a severely reduced binding of NBR1 to p62 in our microscopy-based pull-down assay (Fig. 2f). We also deleted the C-terminal UBA domain (NBR1ΔUBA),

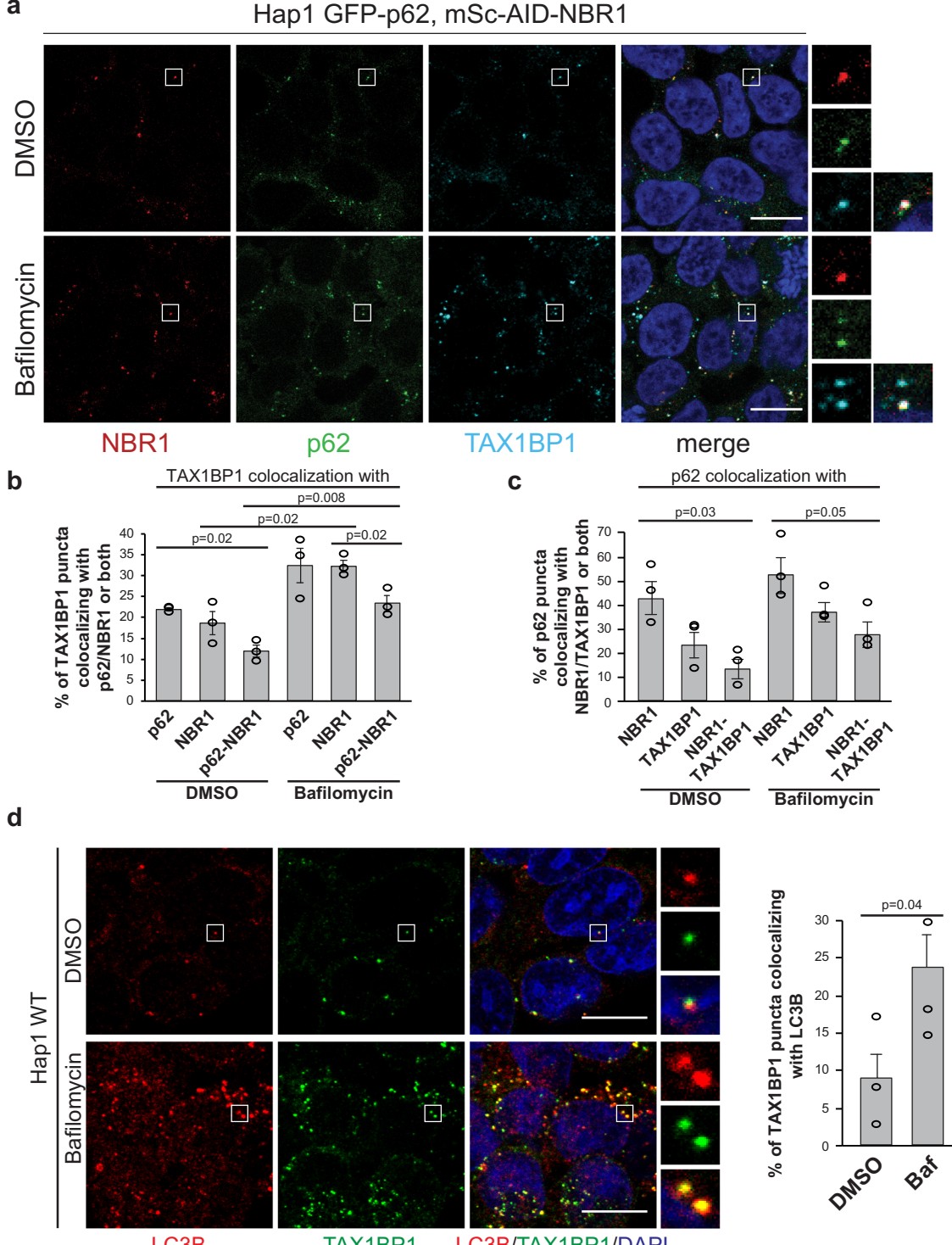

**Fig. 1 p62, NBR1, and TAX1BP1 colocalize in ubiquitin-containing condensates. a** The cargo receptors NBR1 and p62 were endogenously tagged with mScarlet-AID and GFP tags respectively, using CRISPR (Supplementary Fig. 1b). Cells were left untreated (DMSO) or treated with bafilomycin (400 nM for 2 h) and fixed. After fixation, NBR1 and p62 were detected using their endogenous fluorescent tags, while TAX1BP1 was detected by immunofluorescent staining. Scale bar, 10 μm. Validation of endogenous protein tagging in the cell line used for the experiment is shown in Supplementary Fig. 1c and d. **b**, **c** Colocalization of TAX1BP1 with p62, NBR1 or both (**b**) and colocalization of p62 with TAX1BP1, NBR1 or both (**c**), based on the experiments in Fig. 1a. Colocalization analysis was performed with ImageJ. Average percentage of colocalization and SEM for three independent experiments are plotted. An unpaired, two-tailed Student's *t* test was used to estimate significance. *P* values are indicated in the figure. **d** Colocalization of TAX1BP1 with LC3B in HAP1 WT cells mock-treated with DMSO or treated with bafilomycin (400 nM) for 2 h. LC3B and TAX1BP1 were detected by immunofluorescence staining. Scale bar, 10 μm. For the colocalization analysis, average percentages of colocalization ± SEM for *n* = 3 are plotted. An unpaired, two-tailed Student's *t* test was used to estimate significance. *P* values are indicated in the figure.

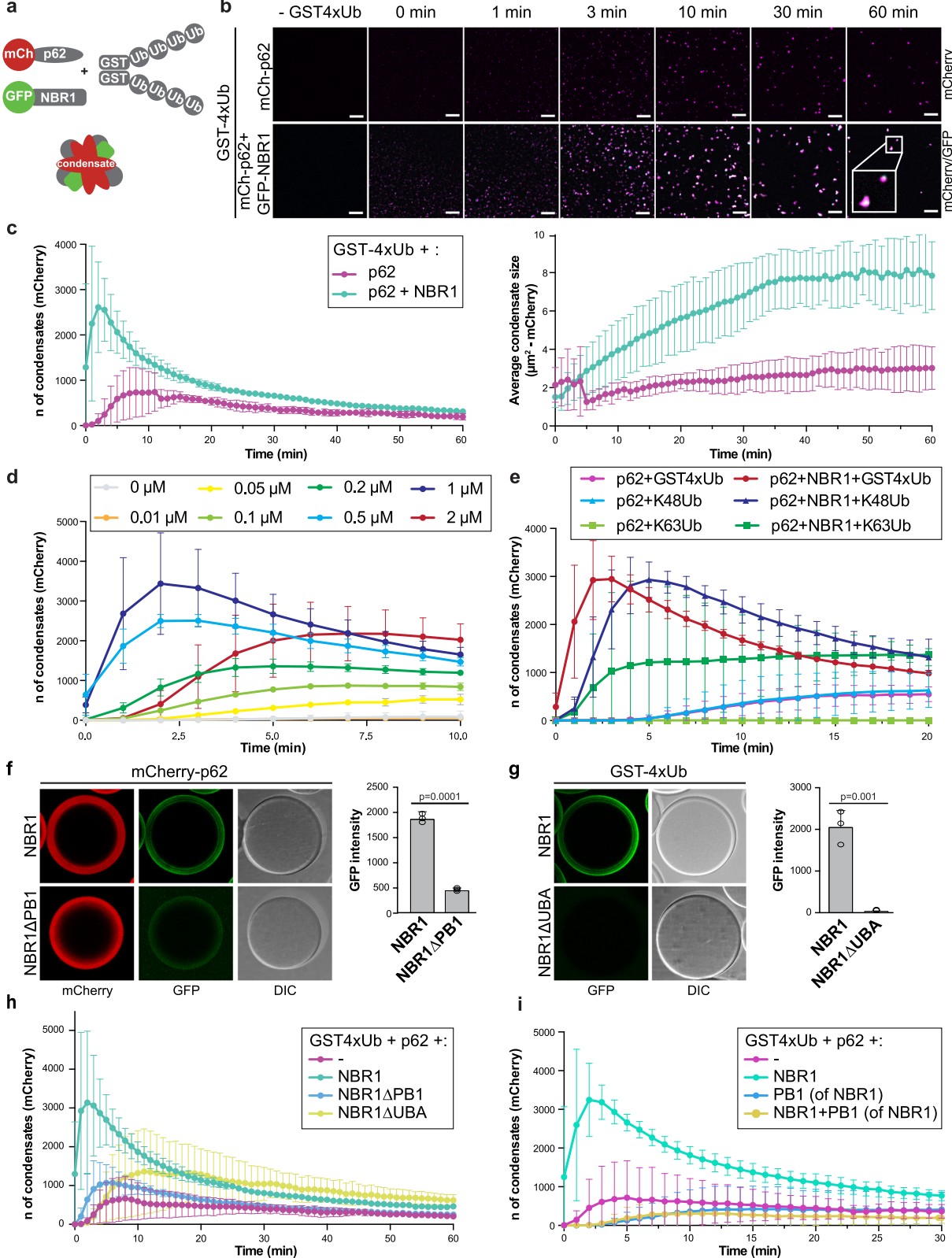

which was previously shown to mediate binding to ubiquitin with a higher affinity than the UBA domain of p62, at least in the context of monoubiquitin, as well as K48- and K63-linked di-ubiquitin[27,31–34]. In the context of the recombinant protein, this deletion also abolished ubiquitin-binding entirely (Fig. 2g). When tested in the condensate formation assay, the NBR1ΔPB1 mutant displayed a severely reduced condensate promoting activity

(Fig. 2h and Supplementary Fig. 2f, g). The PB1 domain alone did not promote condensation and in fact showed a dominant-negative effect on the reaction (Fig. 2i and Supplementary Fig. 2h). This suggests that the interaction of the NBR1 PB1 domain with p62 is not sufficient for the promotion of condensate formation but that this interaction mediates the recruitment of another biochemical activity of NBR1 to the p62

**Fig. 2 NBR1 modulates p62–ubiquitin condensates formation in vitro.** Source data for the assays in this figure are provided as a Source Data file. **a** Schematic representation of the condensate formation assay. **b** Condensate formation assay with GST-4xUb (5 µM), mCherry-p62 (2 µM) and GFP-NBR1 (2 µM). Condensate formation over time in the mCherry and GFP channels was followed by spinning disk microscopy. Scale bar, 15 µm. **c** Quantification of the experiment in (**b**). An average of the number (left) and the size (right) of the condensates formed in the mCherry channel were measured with ImageJ and plotted with standard deviations against time for $n = 3$. The number of condensates formed in the GFP channel and quantification of the control experiment are shown in Supplementary Fig. 2a. A Coomassie stained SDS-Page gel of the proteins used for the assay is shown in Supplementary Fig. 2b. **d** Titration of GFP-NBR1 (concentration range: 0–2 µM) in a condensate formation assay containing 5 µM GST-4xUb and 2 µM mCherry-p62. The average number of mCherry-p62 condensates per field of imaging for $n = 3$ and standard deviations are plotted against time. Coomassie stained SDS-Page gel with protein inputs is shown in Supplementary Fig. 2c. **e** Quantification of a condensate formation assay performed with 5 µM GST-4xUb or in vitro synthesized K48- and K63-linked ubiquitin chains of various length (Supplementary Fig. 2d), incubated with mCherry-p62 (2 µM), with or without GFP-NBR1 (2 µM). BSA was added as a crowding agent to a final concentration of 2%. The average number of mCherry-p62 condensates for $n = 3$ and standard deviations are plotted against time. **f** Microscopy-based pull-down showing the recruitment of GFP-NBR1 (WT or ΔPB1 – 2 µM) to mCherry-p62 coated RFP-trap beads. Average GFP signal intensities and standard deviations for $n = 3$ are shown. An unpaired, two-tailed Student's t test was used to estimate significance. $P$ values are indicated in the figure. Purified NBR1 variants used for this experiment are shown in Supplementary Fig. 2e. **g** Microscopy-based pull-down showing the recruitment of GFP-NBR1 (WT or ΔUBA – 2 µM) to GST-4xUb coated glutathione beads. Average GFP signal intensities and standard deviations for $n = 3$ are shown. An unpaired, two-tailed Student's t test was used to estimate significance. $P$ values are indicated in the figure. Purified NBR1 variants used for this experiment are shown in Supplementary Fig. 2e. **h** Condensate formation assay was performed with NBR1 WT, ΔPB1, and ΔUBA variants (2 µM), mCherry-p62 (2 µM) and GST-4xUb (5 µM). The average number of mCherry-p62 condensates and standard deviations for $n = 3$ are plotted against time. Representative images of the GFP and mCherry channels at the 60 min time point are shown in Supplementary Fig. 2g. SDS-Page gels of purified recombinant GFP-NBR1 (WT and mutants) and the condensate formation assay protein inputs are shown in Supplementary Fig. 2e and f. **i** Condensate formation assay was performed with mCherry-p62 (2 µM) and GST-4xUb (5 µM) with the addition of GFP-NBR1 (2 µM) and/or its isolated PB1 domain (2 µM). The average number of mCherry-p62 condensates and standard deviations for $n = 3$ are plotted against time. SDS-Page gel with the protein inputs for the condensates assay is shown in Supplementary Fig. 2h.

filaments. In line with this, we observed that the UBA domain deletion mutant of NBR1 also showed a severely reduced promoting activity in our assay (Fig. 2h and Supplementary Fig. 2f, g). Both the PB1 and the UBA deletion mutants were still recruited to the condensates (Supplementary Fig. 2g). We, therefore, conclude that NBR1 aids in efficient cargo clustering by p62, by bringing its high-affinity UBA domain to the p62 filaments via its PB1 domain. Consistently, when the p62 UBA domain was replaced with the NBR1 UBA domain the resulting chimeric protein was more efficient than wild-type p62 in forming condensates. In addition, the condensate formation activity of the chimera was not stimulated by the addition of NBR1 (Supplementary Fig. 2i).

**The PB1 and UBA domains of NBR1 are required to promote p62 condensate formation in cells.** In order to compare our in vitro results regarding the promotion of p62-positive condensate formation by NBR1 to the naturally occurring process in cells, we made use of the AID tag attached to endogenous NBR1 in our cell line with endogenously tagged p62 and NBR1 (Supplementary Fig. 1b). The fluorophores on p62 and NBR1 allowed us, for the first time, to image their dynamics at endogenous expression levels in live cells (Fig. 3a). mScarlet retains some stability under acidic conditions and we observed a band positive for mScarlet at ~30 kDa which we interpret as the cleaved form of the fluorophore after NBR1 has been degraded within the lysosome (Supplementary Fig. 1d). Treatment with bafilomycin, which blocks the acidification of the lysosome, led to stabilization of full length mScarlet-AID-NBR1 and a less prominent free mScarlet band (Supplementary Fig. 1d). Live imaging of the distribution of GFP-p62 and mScarlet-AID-NBR1 was also complicated by the pH resistant mScarlet signal emitted from the lysosomes (Fig. 3a). We therefore excluded the mScarlet signal which overlapped with acidified vesicles, stained by LysoTracker Blue and considered only the signal which was outside of these compartments. We quantified the total number of mScarlet-NBR1 particles outside of lysosomes and the number of these particles which colocalized with p62. A large population of NBR1 puncta did not overlap with p62. Upon treatment with wortmannin, which blocks autophagosome formation, we observed a

higher degree of colocalization of the two proteins, suggesting that the double-positive condensates are specifically turned over by autophagy. Consistently, treatment with bafilomycin showed an increase in the population of NBR1 colocalizing with p62 (Fig. 3a).

Having established the cell line and conditions for the live imaging of NBR1 and p62, we stably introduced TIR1, an E3 ligase which upon stimulation with 1-NAA (1-naphtaleneacetic acid) is able to recognize the AID tag fused to NBR1 and induce its proteasomal degradation[30] (Supplementary Fig. 1b). The NBR1 protein in the parental cell line in which no TIR1 was introduced was not affected by treatment with 1-NAA (Supplementary Fig. 3a). The TIR1-containing cell line, however, was able to deplete the majority of the NBR1 protein efficiently within 3 h of treatment (Supplementary Fig. 3a). Despite being a prominent interactor of NBR1 in vivo, p62 was not collaterally degraded by the utilized degron system upon the addition of 1-NAA (Supplementary Fig. 3a).

Next, we examined the effects of acute depletion of NBR1 on the number of p62 condensates. We treated the cell line containing GFP-p62, mScarlet-AID-NBR1, and TIR1 with 1-NAA for 3 h and subjected the cells to live-cell imaging. We observed that the number of GFP-p62 puncta was significantly reduced (Fig. 3b and Supplementary Fig. 3b). Treatment with wortmannin and puromycin promoted higher numbers of total p62 puncta, however, still showed a lower amount of p62 puncta compared to when the 1-NAA treatment was omitted (Fig. 3c). In conclusion, a rapid, targeted depletion of endogenous NBR1 from cells led to a significantly reduced number of p62 puncta, indicating a potentially diminished capacity to cluster cargo for selective autophagy.

We went on to ask if the expression of NBR1 or its PB1 and UBA domain mutants could rescue the depletion of NBR1. To this end, we generated stable cell lines expressing doxycycline-inducible 3xFLAG-iRFP-NBR1 in the HAP1 GFP-p62, mScarlet-AID-NBR1, TIR1 background (Supplementary Figs. 1b and 3c). The NBR1 variants integrated into the cells were either wild-type NBR1 (Supplementary Fig. 3d), a D50R mutant (Fig. 3d), which shows reduced binding to p62[25,27] or an F929A mutant (Fig. 3e), which is defective in ubiquitin binding[31]. Stable clones were

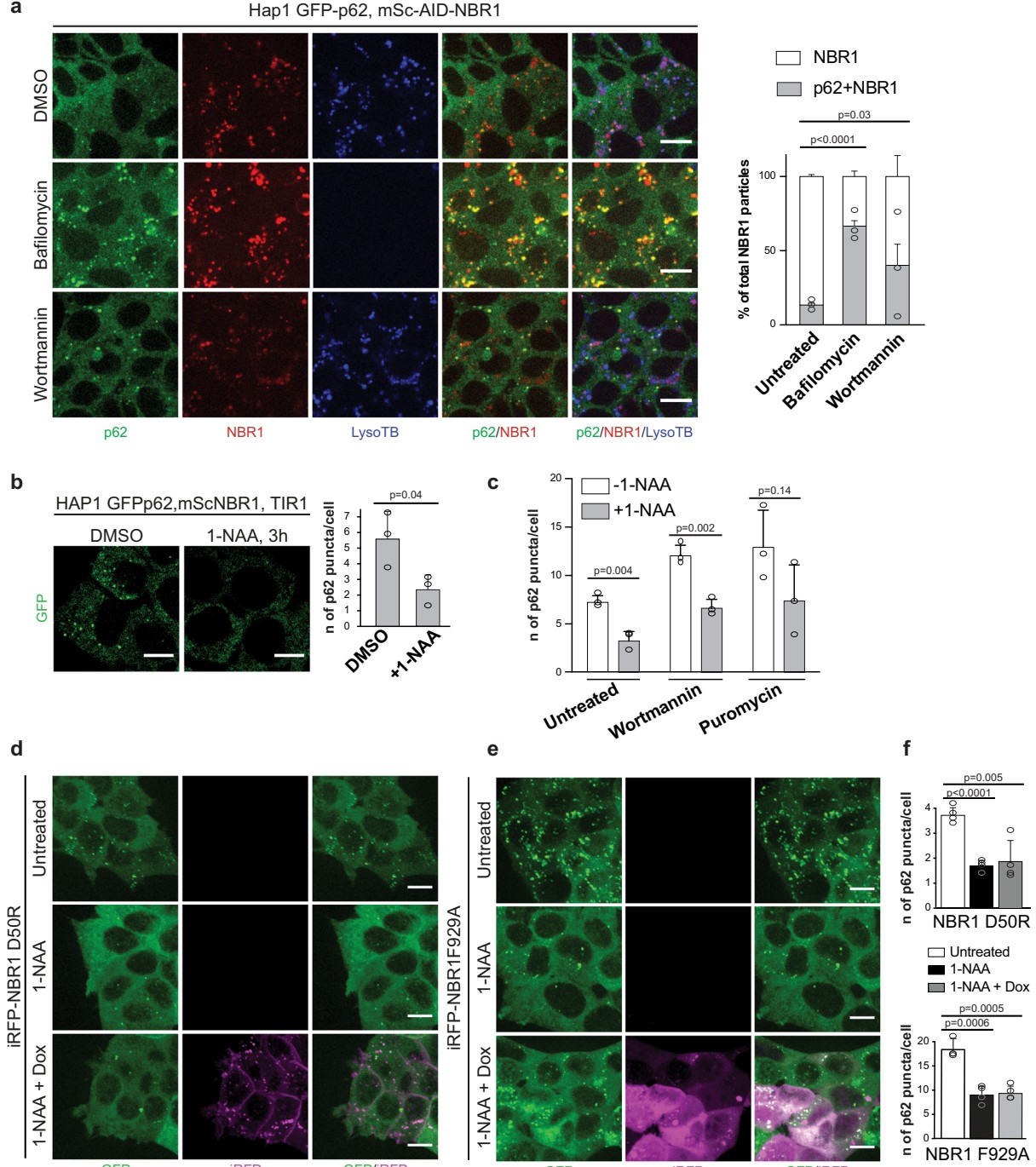

selected based on similar expression levels of iRFP-NBR1 variants after doxycycline treatment (Supplementary Fig. 3c). Despite of not observing any changes of the p62 levels in the cell lines between the clones on a western blot level (Supplementary Fig. 3c), the baseline levels of p62 puncta between the clones differed. In particular, the F929A cell line presented more p62 puncta than the wild-type or D50R clones (Fig. 3d–f and Supplementary Fig. 3d). Taking this into consideration, we compared the number of p62 puncta within each generated cell line at resting state and upon NBR1-depletion as well as doxycycline re-expression. Overexpression of wild-type NBR1 rescued depletion of endogenous NBR1 by 1-NAA as we observed no significant difference between untreated cells and cells rescued with wild-type NBR1 in terms of p62 puncta number

(Supplementary Fig. 3d). The area of the puncta, however, appeared to be larger when NBR1 was overexpressed (Supplementary Fig. 3d), consistent with a previous report[28]. Performing the same treatments for the PB1-mutated D50R cell line we observed a different pattern of NBR1 re-expression (Fig. 3d). The signal for iRFP-NBR1 D50R was much more diffuse and the formed puncta were smaller and less prominent. Quantification showed that, in contrast to the wild-type protein, the over-expression of the D50R mutant of NBR1 was not able to rescue the number of p62-puncta compared to pre-depletion of endogenous NBR1 (Fig. 3f—top graph, Supplementary Fig. 3d). The UBA-mutant F929A cell line, similar to the PB1-mutant cell line, was not able to rescue the p62-puncta phenotype upon 1-NAA-mediated depletion of endogenous NBR1 and mutant re-

**Fig. 3 The PB1 and UBA domains of NBR1 promote condensates formation in cells. a** HAP1 mSc-AID-NBR1, GFP-p62 cells were left untreated (DMSO) or treated with bafilomycin (400 nM) or wortmannin (1 μM) for 2h. Endogenously tagged proteins and LysoTracker Blue (LysoTB) stained lysosomes were visualized by live-cell imaging using a Live Spinning Disk microscope. Scale bar, 10 μm. The total number of NBR1 particles not overlapping with the LysoTB signal are plotted together with the subset of NBR1 particles which colocalize with GFP-p62. Average percentages of colocalization and standard deviations for *n* = 3 are shown. An unpaired, two-tailed Student's *t* test was used to estimate significance. *P* values are indicated in the figure. **b** HAP1 GFP-p62, mSc-AID-NBR1, TIR1 cell line was left untreated (DMSO) or treated with 1-NAA for 3 h. p62 puncta in the cells (GFP channel) were visualized by Live Spinning Disk microscopy. Scale bar, 10 μm. The average number of GFP-p62 puncta/cell and standard deviations for *n* = 3 are shown. An unpaired, two-tailed Student's *t* test was used to estimate significance. *P* values are indicated in the figure. The same experiment performed on the control cell line not containing TIR1 is shown in Supplementary Fig. 3b. **c** HAP1 mSc-AID-NBR1, GFP-p62, TIR1 cells were left untreated (DMSO) or treated with 1-NAA for 3 h in combination with puromycin (5 μg/ml) or wortmannin (1 μM). GFP-p62 puncta in the cells were visualized by live-cell imaging. Average p62 puncta number and standard deviation for *n* = 3 are plotted. An unpaired, two-tailed Student's *t* test was used to estimate significance. *P* values are indicated in the figure. **d**, **e** HAP1 mSc-AID-NBR1, GFP-p62, TIR1 stably expressing iRFP-NBR1 D50R (**d**) or iRFP-NBR1 F929A (**e**) were left untreated (DMSO) or treated with 500 μM 1-NAA with or without 50 ng/ml doxycyclin for 12 h. After treatment GFP-p62 and iRFP-NBR1 puncta were visualized by live-cell imaging. Scale bar, 10 μm. Expression levels of p62 and NBR1 upon treatments are shown in Supplementary Fig. 3c. The same experiment, performed with the GFP-p62, mSc-NBR1, TIR1 cell line stably transfected with iRFP-NBR1 WT, is shown in Supplementary Fig. 3d. **f** Quantification of the experiments in (**d**) and (**e**). The average number of GFP-p62 puncta/cell and standard deviations for *n* = 4 are shown (top graph for the iRFP-NBR1 D50R expressing cells, bottom graph for the NBR1F929A expressing cells). An unpaired, two-tailed Student's *t* test was used to estimate significance. *P* values are indicated in the figure.

expression. The overexpressed iRFP-NBR1 F929A protein showed a diffuse pattern of signal with very few large puncta (Fig. 3e). The levels of p62 puncta were fewer than the pre-treatment levels and similar to when there was no doxycycline-mediated re-expression (Fig. 3f—bottom graph).

We conclude that the PB1 and UBA domains of NBR1 are required to promote optimal p62 condensate formation in vitro and in cells.

**NBR1 directly recruits TAX1BP1 to p62–ubiquitin condensates**. Since TAX1BP1 colocalizes with p62 and NBR1 in condensates in cells (Fig. 1a) and because TAX1BP was recently shown to interact with NBR1[13], we asked if it is recruited to p62–ubiquitin condensates and what its role in their formation might be. We first tested for a direct interaction of TAX1BP1 with p62 and NBR1. TAX1BP1 interacted with both proteins but the binding to NBR1 was notably stronger (Fig. 4a and Supplementary Fig. 4a, b). As the interaction of p62 with TAX1BP1 is weak and given that NBR1 could bind to both p62 and TAX1BP1, we asked if NBR1 could enhance the interaction of p62 with TAX1BP1 by bridging the two molecules. To test this, we immobilized GST-TAX1BP1 on glutathione beads and added mCherry-p62 (Fig. 4b). Thirty minutes thereafter, we supplemented the reaction with GFP-NBR1. Starting from a low level, the signal of mCherry-p62 at the GST-TAX1BP1 beads increased concomitantly with the signal of GFP-NBR1 (Fig. 4b, c). The signal of GFP-NBR1, but not mCherry-p62 alone also increased over time and this increase for GFP-NBR1 was even higher in the absence of mCherry-p62 (Fig. 4c), likely because the mCherry-p62 on the TAX1BP1 beads sterically excludes some GFP-NBR1. These results suggest that NBR1 can bridge p62 and TAX1BP1. We went on to dissect which domain of NBR1 would be responsible for TAX1BP1 recruitment. To this end we used NBR1 fragments covering the majority of its sequence (Supplementary Fig. 4c) and tested their interaction with GST-TAX1BP1 immobilized on glutathione beads (Supplementary Fig. 4d, e). The fragment of NBR1 spanning the CC1 and FW domains (aa 257-498) showed robust recruitment to the beads and the FW domain alone was sufficient for TAX1BP1 binding (Supplementary Fig. 4d). The more robust recruitment of the CC1-FW fragment, compared to the FW domain alone may be due to CC1-mediated oligomerization, which brings more GFP moieties to the beads, although a direct contribution of the CC1 domain to the binding cannot be excluded. However, due to difficulties in obtaining a recombinant isolated CC1 domain, this could not be tested.

To study the interplay of the cargo receptors in context of p62–ubiquitin condensates, we performed condensate formation assays with all three receptors. GST-4xUb was mixed with GFP-p62, NBR1 and mCherry-TAX1BP1 or various combinations thereof (Fig. 4d and Supplementary Fig. 4f, g). As expected GFP-p62 showed robust condensate formation, which was further enhanced by NBR1 (Fig. 4d, e). The addition of mCherry-TAX1BP1 to GFP-p62 and GST-4xUb did not stimulate condensation but rather decreased the number of condensates formed, likely due to a competitive interaction with free GST-4xUB (Fig. 4d, e). mCherry-TAX1BP1 did not show any condensate formation, even though the protein was able to bind to GST-4xUb, albeit weaker than p62 (Fig. 4d, e and Supplementary Fig. 4h, i). Very few condensates were formed when mCherry-TAX1BP1 and NBR1 were added to GST-4xUB in the absence of GFP-62. In contrast, when all three receptors were present, a robust condensate formation reaction was observed and importantly, GFP-p62 and mCherry-TAX1BP1 co-localized in the condensates (Fig. 4d, e). We therefore conclude that apart from directly promoting p62–ubiquitin condensate formation via its PB1 and UBA domains, NBR1 also bridges p62 and TAX1BP1 to recruit the latter to these condensates.

**TAX1BP1 promotes the recruitment of FIP200 to p62–ubiquitin condensates**. Since TAX1BP1, unlike p62 and NBR1, did not directly participate in the formation of ubiquitin condensates (Fig. 4d, e) we asked what its role in the autophagic turnover of these condensates may be. We have recently shown that p62 recruits FIP200 to ubiquitin condensates to induce local phagophore formation around them[10]. TAX1BP1 was also shown to interact with FIP200[7,13]. We therefore directly compared the interaction of the three cargo receptors with FIP200. To this end we coupled GST-FIP200 to beads and added mCherry-p62 or mCherry-TAX1BP1 (Fig. 5a and Supplementary Fig. 5a). p62 detectably bound to FIP200 and this interaction was enhanced by 4 phospho-mimicking mutations, as we reported earlier[10]. Compared to p62, or its phospho-mimicking mutant, TAX1BP1 bound notably stronger to the GST-FIP200 beads (Fig. 5a). Surprisingly, we found that also NBR1 could directly interact with GST-FIP200 and, under the conditions tested, it bound even stronger to FIP200 than TAX1BP1 (Fig. 5b and Supplementary Fig. 5b). Using NBR1 fragments (Supplementary Fig. 5c) and the C-terminal region (CTR) of FIP200 we mapped NBR1-FIP200 interaction to a region of NBR1 containing its CC2 domain (Supplementary Fig 5d, e). In particular, we found this interaction

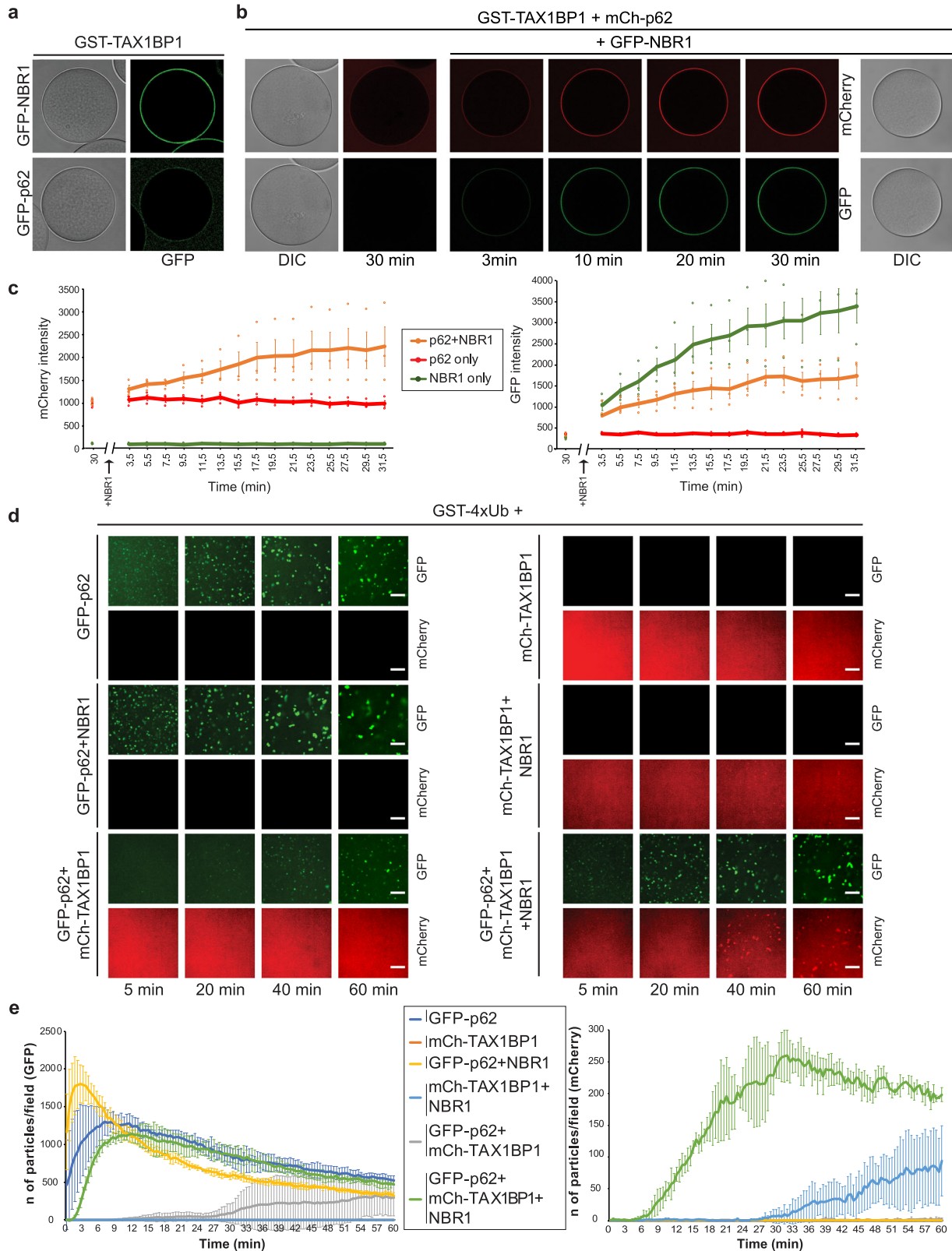

to take place between the CC2 domain of NBR1 and the Claw domain of FIP200 (Fig. 5c and Supplementary Fig. 5f). Notably, the same residue in the FIP200 Claw domain which contributes to p62 binding (R1573)[10], plays an important role in the interaction with NBR1. In fact, a R1573D mutant of FIP200's Claw domain, failed to bind full-length NBR1 or its CC2 domain (Fig. 5c). We

further found that FIP200 co-immunoprecipitated with endogenously tagged GFP-NBR1 from HAP1 cell lysates (Fig. 5d).

We next tested the contributions of p62, NBR1, and TAX1BP1 in the recruitment of FIP200 to ubiquitin condensates (Fig. 5e, f and Supplementary Fig. 5g). We found that full-length FIP200-GFP was weakly recruited to p62–ubiquitin condensates. The addition of NBR1 increased the FIP200-GFP signal at the

**Fig. 4 NBR1 recruits TAX1BP1 to p62–ubiquitin condensates in vitro. a** Microscopy-based pull-down showing TAX1BP1 interaction with NBR1 and p62. The recruitment of GFP-NBR1 (2 µM) and GFP-p62 (2 µM) to GST-TAX1BP1 coupled glutathione beads at the equilibrium was followed by confocal microscopy. The experiment was done in three independent replicates. Negative binding control with GST coupled beads is shown in Supplementary Fig. 4a. SDS-Page gel with the protein inputs is shown in Supplementary Fig. 4b. **b** GST-TAX1BP1 coupled glutathione beads were incubated with 2 µM mCherry-p62 for 30 min and imaged by confocal fluorescent microscopy. Then, GFP-NBR1 (final concentration 2 µM) was added to the reaction and the recruitment of both mCherry-p62 and GFP-NBR1 to GST-TAX1BP1 coated beads was followed by confocal fluorescent microscopy for 30 min (one image every 2 min). **c** Quantification of the experiment shown in (**b**). The mCherry and GFP signal intensity on the beads is plotted against time. A reaction without GFP-NBR1 and one without mCherry-p62 were used as controls to monitor the recruitment of the single receptors over time. Average signal intensity (mCherry—left graph and GFP—right graph) and standard deviations for $n = 3$ are shown. **d** Dynamics of ubiquitin condensate formation with GST-4xUb (5 µM), GFP-p62 (2 µM), NBR1 (1 µM) and mCherry-TAX1BP1 (2 µM). Cargo receptors were added to the reactions in the indicated combinations. Three independent replicates of the experiment were performed. Images of the GFP-p62 and mCherry-TAX1BP1 condensates for every condition at representative time points are shown. Scale bar: 10 µm. Negative controls for the condensate formation reactions and protein inputs are shown in Supplementary Fig. 4c and d respectively. **e** Quantification of the condensate formation experiment in (**d**). The number of condensates per field of imaging at every time point in each channel were counted and plotted against time. The number of p62 condensates (green channel) is shown on the left graph and the number of TAX1BP1 condensates (red channel) is shown on the right graph. Average fluorescence intensity and standard deviation for $n = 3$ is shown. Source Data are provided as a Source Data file.

condensates. The strongest recruitment of FIP200 was observed when TAX1BP1 was added to the reaction (Fig. 5e, f). Thus, a major function of TAX1BP1 could be the recruitment of FIP200 to NBR1 positive p62–ubiquitin condensates.

Moving forward, we tested the interplay of p62, NBR1, and TAX1BP1 in cells by knocking down NBR1 and assessing the recruitment of TAX1BP1 to p62 condensates (Fig. 6a and Supplementary Fig. 6a). Upon depletion of NBR1 the colocalization of TAX1BP1 and p62 dropped significantly (Fig. 6a), consistent with the results of our reconstituted system (Fig. 4) and the reported interaction of NBR1 with TAX1BP1[13]. Next, we explored the requirement of TAX1BP1 for the recruitment of FIP200 to p62 condensates. Similar to NDP52, TAX1BP1 interacts with the coiled-coil domain of FIP200, while p62 and NBR1 bind the C-terminal Claw domain of FIP200 (Fig. 5c)[7,10,35]. The Claw mediated interaction with p62 and NBR1 is abolished by the R1573D mutation (Fig. 5c)[10]. To compare the modes of recruitment of FIP200 to the p62–ubiquitin condensates, we introduced the R1573D mutation into the endogenous FIP200.

The point mutation did not affect FIP200 expression levels, but resulted in an increased level of the p62 protein and reduced LC3B lipidation (Supplementary Fig. 6b, c), suggesting that Claw domain-mediated recruitment of FIP200 to cargo material is responsible for a considerable fraction of the basal autophagic activity in the cells. We then compared the recruitment of FIP200 to p62 condensates in wild-type and R1753D cells and upon TAX1BP1 depletion by siRNA (Fig. 6b, c and Supplementary Fig. 6c). TAX1BP1 depletion resulted in about 40% reduction of p62–FIP200 colocalization and an overall reduction of FIP200 puncta. In the R1753D cells, the number of FIP200 puncta and the degree of colocalization with p62 was further decreased, which dropped even more upon depletion of TAX1BP1. The same trend was observed when we blocked autophagosome formation with the PI3K inhibitor wortmannin (Fig. 6c). Although TAX1BP1 knockdown did not dramatically affect the overall level of p62 protein (Supplementary Fig. 6c), it led to an increase of p62 puncta in cells (Supplementary Fig. 6d). Thus, TAX1BP1 and Claw domain-mediated receptor interaction cooperate in the autophagic turnover of p62–ubiquitin condensates.

## Discussion

Here we employed reconstituted systems in combination with cell biological assays to dissect the interplay of the p62, NBR1 and TAX1BP1 cargo receptors in the formation and autophagic degradation of ubiquitin condensates, a central process in cellular proteostasis[15]. Reconstituted assays offer full control over the

components studied and therefore enable the discovery of functional interaction, which are difficult to study in cells. We found that while all three receptors have overlapping biochemical activities, each factor has its own specialty. In particular, p62 is the main driver of condensate formation, consistent with previous findings[19,20]. NBR1 directly enhances condensation via the provision of a high-affinity UBA domain to the p62-NBR1 heterooligomer. In addition, NBR1 recruits TAX1BP1 to the p62–ubiquitin condensates and TAX1BP1 in turn is the major recruiter of FIP200 to the condensates (Fig. 7).

It was previously shown that NBR1 enhances p62–ubiquitin condensate formation[20,24,27,28]. Here we show that the PB1 and UBA domains directly impact the ability of p62 to form condensates with ubiquitinated proteins. Our results suggest that the PB1 mediated recruitment of NBR1 to p62 is required to promote condensate formation. Nevertheless, the PB1 domain of NBR1 itself is not sufficient to promote condensate formation and even inhibits the reaction when added on top of wild-type NBR1. This suggests that NBR1 harbors other biochemical properties which upon recruitment to p62 promote condensation. Our results further show that the NBR1 UBA domain confers one of these activities as deletion of the UBA severely reduced the condensation promoting effect. It appears that the UBA domain of NBR1 must be connected to the p62 filaments via the PB1—PB1 domain interaction in order to promote condensation because the PB1 mutant of NBR1 was still recruited to the condensates, possibly via the UBA domain-mediated interaction with ubiquitin but nevertheless showed reduced facilitation of condensation. The UBA domain of NBR1 has a higher affinity for ubiquitin compared to the UBA of p62[31,33] and thus the heterooligomeric NBR1-p62 complex may have a higher affinity for ubiquitinated substrates than homooligomeric p62 filaments. Interestingly, at least in vitro, the condensation promoting activity of the NBR1 was not completely lost upon deletion of the NBR1 UBA domain. NBR1 itself dimerizes or oligomerizes[20,27]. It is therefore conceivable that NBR1 caps and shortens p62 filaments via its PB1 domain[24] but that the heterooligomers have a non-filamentous shape that is more efficient in condensate formation.

In addition to promoting condensate formation, NBR1 also directly promotes their autophagic degradation. Firstly, NBR1 binds to the positively charged R1573 in the FIP200 Claw domain via its highly negatively charged CC2 domain and secondly, it interacts with TAX1BP1[13]. Our data show that in vitro and in cells, the recruitment of FIP200 via the Claw domain, which binds p62 and NBR1, and the recruitment by TAX1BP1 cooperate for robust FIP200 recruitment for autophagosome formation.

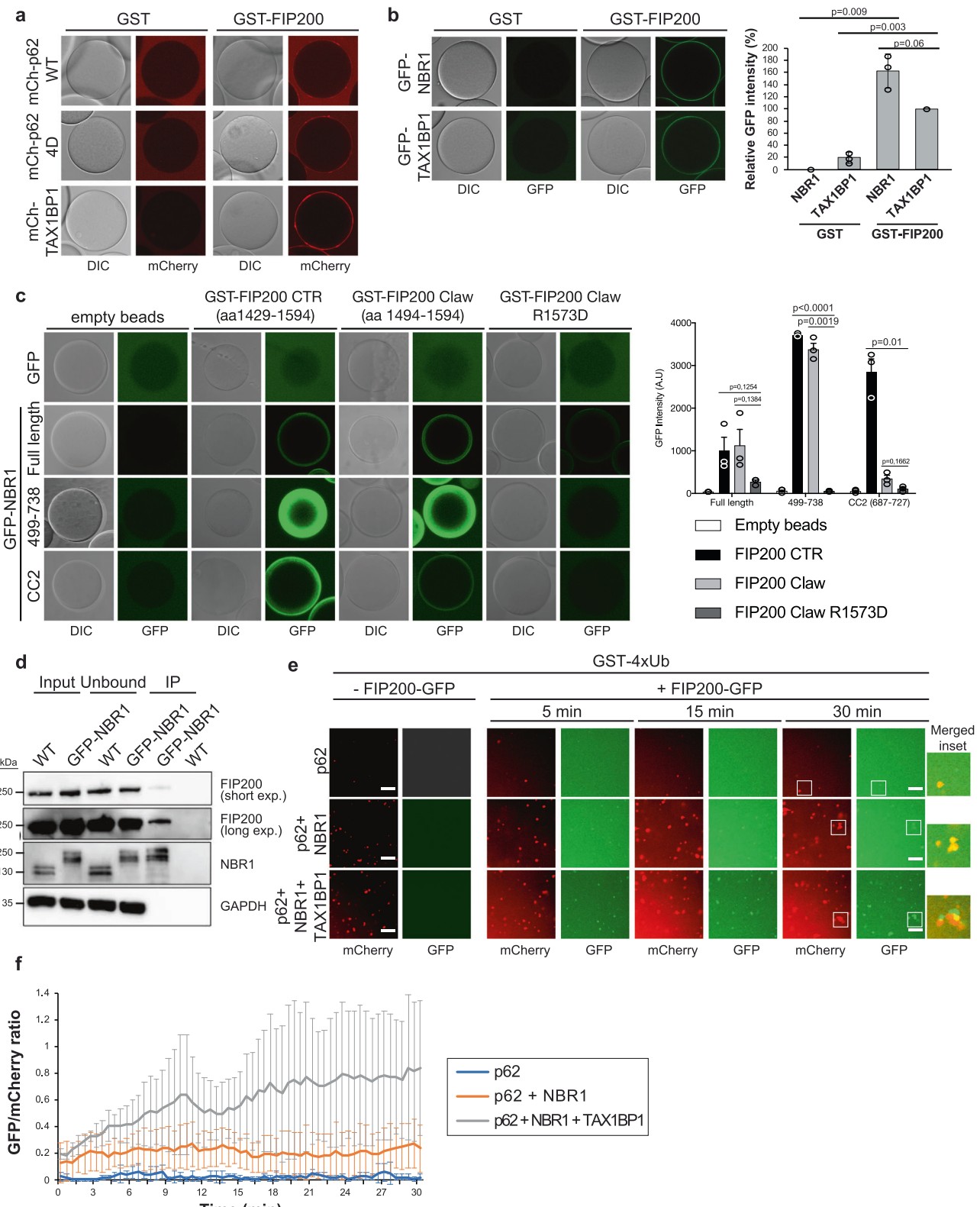

Why are at least three autophagy receptors required for the autophagy of ubiquitin-containing condensates? The reason may be that this process entails the condensation reaction prior to the subsequent induction of autophagosome formation in vicinity of the condensates by the recruitment of the autophagy machinery. This is markedly different when compared to other types of selective autophagy mediated by optineurin and NDP52 such as

mitophagy or xenophagy where the bulky cargo material does not require condensation[1]. During condensation, the cargo is sequestered and this process may also occur at sites that are not permissive for autophagosome formation, in particular in cells with more complex shapes such as neurons. Indeed, autophagosome formation is only possible in the presence of a membrane source. Thus, efficient TAX1BP1 and subsequent FIP200

**Fig. 5 Receptors mediate the recruitment of FIP200 to ubiquitin condensates. a** Glutathione beads were coupled with GST or GST-FIP200 and incubated with mCherry-p62 WT (2 µM), mCherry-p62 4D (phosphomimicking mutant – 2 µM) or mCherry-TAX1BP1 (2 µM). After 1 h incubation at room temperature beads were imaged at the equilibrium with a LSM700 microscope. The experiment was done in three independent replicates. Recombinant purified proteins used for the assay are shown in Supplementary Fig. 5a. **b** GST or GST-FIP200 coupled glutathione beads were incubated with GFP-NBR1 (2 µM) or GFP-TAX1BP1 (2 µM). After 30 min incubation the beads at the equilibrium were visualized with a LSM700 confocal microscope. The average GFP signal ± SEM for $n = 3$ is plotted. An unpaired, two-tailed Student's $t$ test was used to estimate significance. $P$ values are indicated in the figure. Recombinant purified proteins used in the assay are shown in Supplementary Fig. 5b. **c** Glutathione beads were left empty or coupled with GST-FIP200 CTR (aa 1429–1594), GST-FIP200 Claw (aa 1494–1594), or the GST-FIP200 Claw R1573D mutant. Each set of beads was incubated with 2 µM GFP or 2 µM of the indicated GFP-NBR1 constructs (see also Supplementary Fig. 5c) and imaged at the equilibrium by confocal fluorescent microscopy. The average GFP signal intensity ± SEM for $n = 3$ is shown. An unpaired, two-tailed Student's $t$ test was used to estimate significance. $P$ values are indicated in the figure. SDS-Page gel with pull-down inputs is shown in Supplementary Fig. 5f. **d** Cell lysates from HAP1 WT and GFP-AID-NBR1 (Supplementary Fig. 1c) were incubated with GFP-trap beads. Beads bound material was analyzed by western blot to detect beads bound GFP-AID-NBR1 and co-immuno-precipitated FIP200. Input samples used for the experiment, unbound proteins, and immunoprecipitated proteins (IP) are shown in the figure. The experiment was done in three independent replicates. **e** Condensate formation reaction was performed with GST-4xUb (5 µM) and the indicated combinations of mCherry-p62 (2 µM), mCherry-TAX1BP1 (2 µM) and NBR1 (1 µM). After 30 min incubation at room temperature, formed condensates were imaged by spinning disk microscopy. Then, GFP-FIP200 (5 µM) was added to each reaction and its recruitment to the condensates over time was followed. Images of representative time points are shown. Scale bar: 10 µm. Protein inputs for the assay are shown in Supplementary Fig. 5g. **f** The number of condensates in each channel was counted and the GFP/mCherry ratio was plotted against time. Average GFP/mCherry ratio and standard deviation for $n = 3$ is shown. Source data are provided as a Source Data file.

recruitment may occur in a temporally and spatially regulated manner to couple the efficient recruitment of the autophagy machinery to this membrane. Furthermore, TAX1BP1 may only be recruited after the condensates have matured to contain the right kind and amount of cargo destined for degradation. How its recruitment may be regulated is currently unclear. However, the presence of ubiquitin per se, may be insufficient for TAX1BP1 recruitment (Fig. 4d)[13].

Future work will have to address how cargo collection and condensation are coupled to the recruitment of the autophagy machinery in space and time. Our study provides crucial insights into the mechanisms of action and the division of labor of the p62, NBR1 and TAX1BP1 cargo receptors in these processes, which may allow more targeted interventions in conditions where the degradation of ubiquitinated proteins is compromised, such as in neurodegeneration.

# Methods

**Cell lines and cell culture**. HAP1 WT and FIP200 KO cells were purchased from Horizon Discovery and all the other lines were generated from parental HAP1 WT as described below. All HAP1 cell lines were cultivated at 37 °C in humidified 5% $CO_2$ atmosphere and grown in Iscove´s Modified Dulbecco´s Medium (IMDM— Gibco, Thermo Fisher Scientific) supplemented with 10% fetal bovine serum (FBS —Sigma) and 10% Penicillin-Streptomycin (Sigma).

*Generation of endogenously tagged and mutant cell lines*. HAP1 GFP-p62 cells, used as parental cell line to generate HAP1 GFP-p62, mSc-AID-NBR1 were generated as described previously[20].

To endogenously tag NBR1 with an mScarlet-AID or GFP-AID tags, the genomic area ~1500 bp up and downstream of the ATG start codon was amplified from isolated HAP1 genomic DNA (obtained using GeneJET Genomic DNA Purification Kit, Thermo Fisher Scientific) and sub-cloned into pUC19 vector (Addgene). The corresponding tags, mScarlet-AID and GFP-AID were amplified via PCR and assembled via Gibson cloning (Gibson Assembly Master Mix, NEB). The resulting plasmids were sequenced. The chosen gRNAs were cloned into an all-in-one (AIO) plasmid containing Cas9D10A nickase (Addgene) via standard BbsI cleavage sites and the resulting plasmid was sequenced. The AIO plasmid and the template plasmid were co-transfected into HAP1 GFP-p62 (for mScarlet-AID-NBR1) or HAP1 WT (for GFP-AID-NBR1) cells using FuGene 6 transfection reagent (Promega). Forty-eight after transfection cells were sorted for mScarlet fluorescence in bulk, left to expand until confluent in a 15 cm dish and re-sorted for mScarlet fluorescence as single clones. The sorted single clones were left to expand in 96-well plates. Integration of the tags in the target genomic region was confirmed first by genotyping PCR (with primers annealing outside the homology template around the ATG), by sequencing cDNA generated by reverse transcription of extracted mRNA and western blot (Supplementary Fig. 1b). Clones containing the tags, which had not integrated Cas9 in the genome were selected for further experiments.

Mutagenesis of endogenous FIP200 in HAP1 cells by CRISPR/Cas9 was performed using Cas9 nuclease and a dsDNA repair template containing the R1573D mutation. A guide RNA was designed around the R1573 residue of FIP200 and cloned into pSp-GFP-Cas9 plasmid (Addgene) using the BbsI restriction site. For the dsDNA repair template, a 2 kb homology region spanning 1 kb upstream and downstream of the mutation site was amplified from HAP1 WT cDNA and cloned into pUC19 plasmid (Addgene) using BamHI/NdeI restriction sites. The R1573D mutation was generated using Round the Horn PCR mutagenesis. Then, a second cycle of Round the Horn PCR was used to mutate the PAM sequence and to insert a silent BstNI restriction site in proximity of the mutation, to be used for clone screening. Forty percent confluent HAP1 WT cells were co-transfected with the two plasmids described above (5 µg each) using 30 µl Fugene6 transfection reagent (Promega) in OptiMEM medium. Twenty-four hours after transfection GFP-positive cells were sorted by fluorescence-activated cell sorting (FACS), plated in 10 cm dishes (200,000 cells/dish) and let to grow until confluent. Single cells were then sorted into 96-well plates for clonal selection. Clones were screened by BstNI restriction digestion of a 500 bp PCR product surrounding the mutation site and further validated by sequencing. FIP200 expression levels and the levels of other autophagy-related genes (p62, LC3B) were analyzed by immunoblotting and compared to WT HAP1 cells (Supplementary Fig. 6b).

*Generation of TIR1 and iRFP-NBR1 stable cell lines*. A stable cell line for the depletion of the endogenously tagged mScarlet-AID-NBR1 was generated, using HAP1 GFP-p62, mSc-AID-NBR1 as parental cell line, by introducing a TIR1-9xmyc lentiviral plasmid, carrying a puromycin resistance gene, which integrated randomly in the genome. Cells were transfected using a FuGene 6 transfection reagent (Promega) and selected with puromycin (5 µg/ml) until the control, un-transfected cells died. Cells were then single sorted to 96-well plates and left to expand. Clones were first screened by PCR targeting the inserted TIR1-9xmyc gene and validated by western blot (myc tag detection and NBR1 depletion—Supplementary Fig. 3a). The stable cell lines expressing doxycycline inducible 3xFLAG-iRFP-NBR1 variants were generated in a similar fashion by selection with G-418 (1200 µg/ml), using the HAP1 GFP-p62, mSc-NBR1, TIR1 as parental cell line. They were FACS single-sorted based on iRFP fluorescence after a 24 h treatment with 500 ng/ml doxycycline. After PCR screening, they were treated with various doxycycline concentrations for various periods of time and the expression levels of the FLAG-tag and NBR1 were monitored by western blot. Clones showing similar expression levels of NBR1 variants upon doxycycline treatment were used for further experiments (Supplementary Fig. 3c).

**Protein expression and purification**. mCherry- and GFP-p62 WT, GST-4xUb and GST-FIP200 CTR were expressed and purified as previously described[10,20,26]. The mCherry-p62-NBR1 chimera construct was obtained by replacing p62 UBA domain (aa 339-434) with NBR1 UBA domain (aa 913-966) in the plasmid containing p62 WT. The chimeric protein was expressed and purified as mCherry-p62 WT. The GST-FIP200 Claw/Claw R1573D were expressed and purified as the GST-FIP200 CTR construct. In vitro synthesis of K48- and K63-linked ubiquitin chains was performed as previously described[20].

GFP-NBR1 fragments were obtained by cloning the GFP and the respective NBR1 ORFs in frame with the 6xHis-TEV-tag into pETDuet plasmid. Proteins were expressed in *E. coli* Rosetta (DE3) pLysS cells. Cells were grown at 37 °C to an $OD_{600}$ of 0.6, induced with 0.1 mM IPTG, and grown for additional 16 h at 18 °C. Cells were harvested by centrifugation and the cell pellet was resuspended in lysis

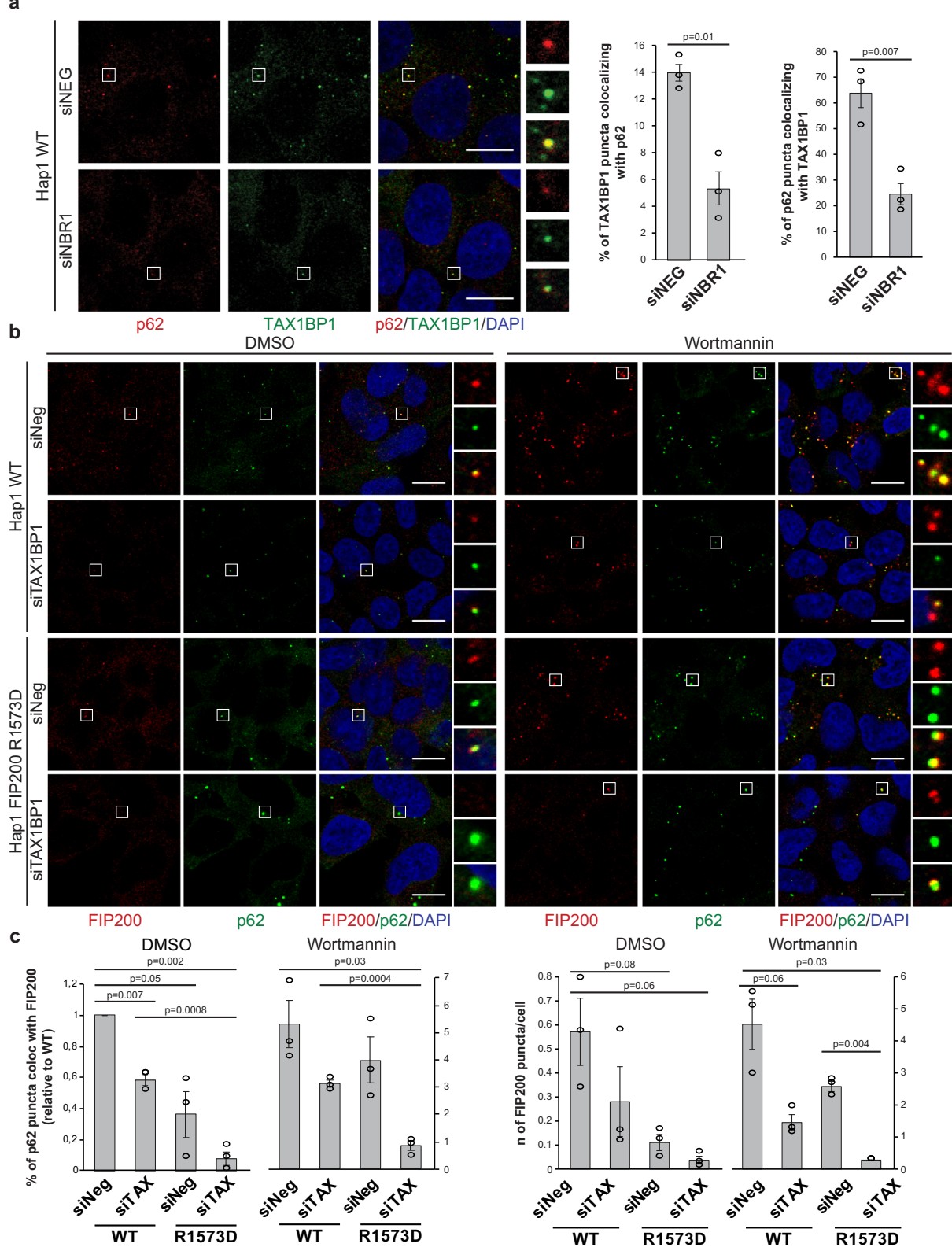

buffer (50 mM HEPES pH 7.5, 300 mM NaCl, 2 mM MgCl₂, 2 mM β-mercaptoethanol, Pefabloc® SC-Protease Inhibitor (Roth), DNAse I (Sigma)). The resuspended pellet was flash frozen and stored at −80 °C. For purification the resuspended pellet was thawn in a room temperature water bath and spun down at 185,000 × *g* for 20 min using a Beckman Ti45 rotor. The supernatant was filtered through a 0.45 μm syringe filter and loaded onto a 5 ml HisTrap column (GE Healthcare). An imidazole stepwise gradient (50–300 mM) was used to elute the proteins which were then concentrated with an appropriate concentrating filter

(Millipore). The constructs were loaded on a Superose 6 or Superdex 200 10/300 column, depending on the size of the purified fragment, and eluted in 25 mM HEPES—pH 7.5, 150 mM NaCl, 1 mM DTT buffer. The eluted fractions were tested on SDS-PAGE, pooled appropriately, concentrated, aliquoted and flash frozen for storage at −80 °C.

For TAX1BP1 constructs, TAX1BP1 ORF was amplified from HeLa cells cDNA and cloned with the respective tags (GST, 10xHis-mCherry and 10xHis-GFP) by standard restriction cloning into pLIB vectors.

**Fig. 6 TAX1BP1 promotes the recruitment of FIP200 to p62 condensates in cells. a** HAP1 WT cells were treated with a non-targeting siRNA (siNeg) or with NBR1 siRNA (20 nM for 48 h). Endogenous p62 and TAX1BP1 were visualized by immunofluorescent staining. Scale bar, 10 μm. TAX1BP1—p62 colocalization was analyzed. Average percentage of colocalization ± SEM for three independent experiments is shown. An unpaired, two-tailed Student's *t* test was used to estimate significance. *P* values are indicated in the figure. Western blot showing the efficiency of the siRNA treatment is shown in Supplementary Fig. 6a. **b** HAP1 WT cells or cells where the R1573D mutation was introduced in the endogenous FIP200 (see also Supplementary Fig. 6b) were treated with non-targeting siRNA (siNeg) or with TAX1BP1 siRNA (20 nM). After 48 h cells were left untreated (DMSO) or treated with wortmannin (1 μM) for 2 h. Endogenous p62 and FIP200 were visualized by immunofluorescent staining. Scale bar, 10 μm. The efficiency of the siRNA treatment and levels of autophagy markers upon treatment are shown in Supplementary Fig. 6c. **c** Analysis of p62 colocalization with FIP200 (left plot) and FIP200 puncta/cells (right plot) for the experiment in Fig. 6b. Average puncta number/percentage of colocalization ± SEM for *n* = 3 are shown. An unpaired, two-tailed Student's *t* test was used to estimate significance. *P* values are indicated in the figure.

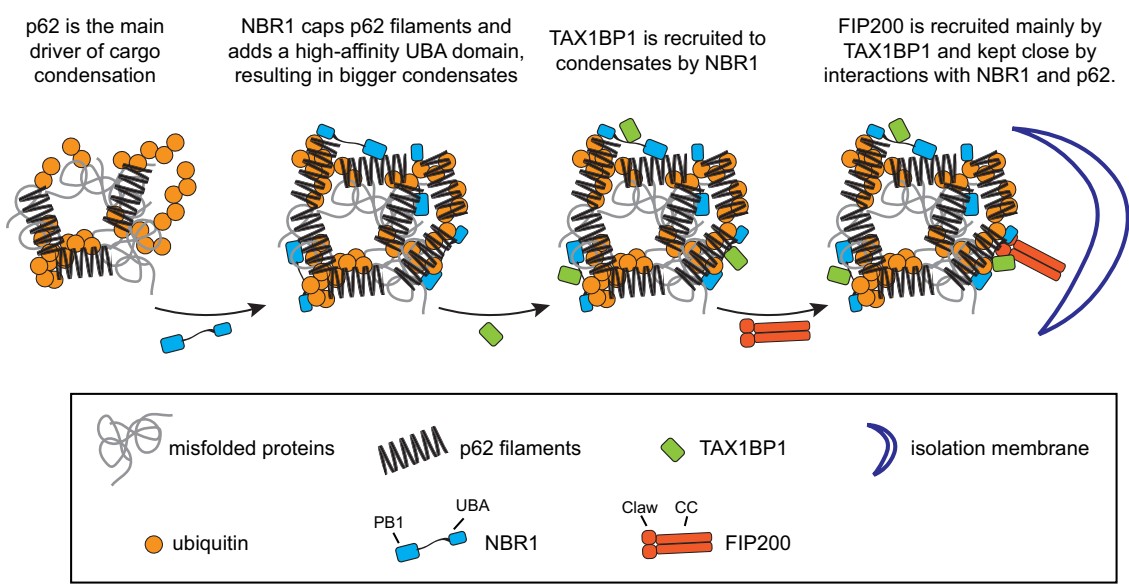

**Fig. 7 Model of cargo receptors' interplay in autophagy initiation.** The model illustrates how each cargo receptor contributes to ubiquitin condensate formation and to the recruitment of the autophagy machinery to initiate autophagy.

To obtain GST-FIP200 and GST-FIP200-GFP constructs the insect codon-optimized FIP200 gene was purchased from GenScript and cloned with the respective tags into pGB-02-03 via the Gibson assembly method by the Vienna BioCenter Core Facilities (VBCF) Protech Facility.

6xHis-NBR1 and 6xHis-Strep-TEV-GFP-NBR1 constructs were generated by cloning the respective tags and the protein ORF, amplified from HeLa cells cDNA, into pcDNA3.1(+). NBR1 ΔPB1 and ΔUBA deletion mutants were obtained by amplification of the plasmid encoding 6xHis-Strep-TEV-GFP-NBR1 with primers that exclude the PB1 or UBA domains respectively. The NBR1 proteins, fused with the respective tags were then subcloned into pFastBac HT B vector. TAX1BP1, FIP200 and NBR1 constructs generated as described above were used for expression in Sf9 insect cells using the Bac-to-Bac system. The bacmid DNAs (2.5 μg per construct), obtained by amplification in DH10BacY cells were used to transfect Sf9 insect cells using FuGene transfection reagent (Promega). About 7 days after transfection the P0 virus was harvested and used to produce a P1 virus stock, which in turn was used to infect 1 L Sf9 cells (1 million/ml) for protein expression. After infection cells were monitored and harvested by centrifugation when they reached a viability of 95–98%. Cell pellets were washed with PBS, flash-frozen in liquid nitrogen, and stored at −80 °C until purification.

*TAX1BP1 constructs purification.* Cell pellets corresponding to 1 L culture were thawed and resuspended in 40 ml lysis buffer (50 mM HEPES—pH 7.5, 300 mM NaCl, 0.5% CHAPS, 1 mM MgCl₂, 1 mM β-mercaptoethanol, Benzonase (5 U/ml), Complete EDTA-free protease inhibitors (Roche)). Cells were lysed with a Dounce homogenizer and sonicated on ice for 1 min, 50% cycle, 50% power. Lysates were cleared by ultracentrifugation at 72,000 × *g* for 45 min at 4 °C, using a Beckman Ti45 rotor. For GST-TAX1BP1 purification, lysate was incubated with 5 ml glutathione beads (GE Healthcare), pre-washed in WBI (50 mM HEPES—pH 7.5, 300 mM NaCl, 1 mM DTT), for 90 min at 4 °C on a tube roller. Beads were washed five times in WBI, 1 time in WBII (50 mM HEPES—pH 7.5, 700 mM NaCl, 1 mM DTT) and two times in WBI. Beads were then incubated with 10 ml elution buffer (WBI + 50 mM glutathione (Sigma)—pH adjusted to 8.0) for 1 h at 4 °C. The eluate was filtered through a 0.45 μm syringe filter and concentrated to a final volume of 500 μl using 100 kDa MWCO concentrators (Millipore), before being

further purified by size exclusion chromatography on a Superose 6 Increase 10/300 column (GE Healthcare) in 25 mM HEPES—pH 7.5, 150 mM NaCl, 1 mM DTT.

For 10xHis-GFP and 10xHis-mCherry TAX1BP1 constructs, proteins were purified by affinity chromatography on HisTrap 5 ml HP column (GE Healthcare) with stepwise elution in imidazole gradient (50–300 mM) followed by size exclusion chromatography on a Superose 6 Increase 10/300 column.

*GST-FIP200 and FIP200-GFP purification.* Cell pellets corresponding to 1 L culture were thawed and resuspended in 40 ml lysis buffer (50 mM HEPES—pH 7.5, 300 mM NaCl, 10% glycerol, 0.5% CHAPS, 5 U/ml Benzonase (Sigma), 1 mM DTT, CIP protease inhibitor (Sigma), cOmplete EDTA-free protease inhibitor cocktail (Roche)). Cells were lysed by Dounce homogenizer followed by 1 min sonication at 50% cycles and 50–60% power. Lysate was clarified by centrifugation at 72,000 × *g* for 45 min at 4 °C using a Beckman Ti45 rotor and the supernatant was incubated with 5 ml pre-equilibrated Glutathione Sepharose 4B beads (GE Healthcare) O/N at 4 °C on a tube roller. Beads were then washed seven times in wash buffer (50 mM HEPES—pH 7.5, 200 mM NaCl, 1 mM MgCl₂, 1 mM DTT and the protein was eluted by incubation with 10 ml elution buffer (50 mM HEPES, 200 mM NaCl, 50 mM glutathione (Sigma), 1 mM DTT—pH adjusted to 8.0) for 2 h at 4 °C on a tube roller. The eluate was filtered through a 0.45 μm syringe filter and concentrated to a final volume of 500 μl using 100 kDa MWCO concentrators (Millipore). The concentrated sample was further purified by size exclusion chromatography on a Superose 6 Increase 10/300 column (GE Healthcare) in 25 mM HEPES—pH 7.5, 200 mM NaCl, 1 mM DTT. For the purification of FIP200-GFP (obtained by cleavage of the GST tag from GST-FIP200-GFP) the elution step was replaced by O/N incubation of the beads bound protein with preScission 3C protease in 10 ml wash buffer.

*NBR1 WT, ΔPB1, and ΔUBA purification.* Cell pellets, corresponding to 500 ml culture were resuspended in 20 ml lysis buffer (50 mM HEPES—pH 7.5, 150 mM NaCl, 2 mM MgCl₂, 0.25% CHAPS, 10 mM Imidazole, 2 mM β-mercaptoethanol, cOmplete EDTA-free protease inhibitor (Roche), 1 mM Pefabloc® SC-Protease Inhibitor (Roth), 5 U/ml Benzonase (Sigma), DNase (Sigma), RNase (Sigma), CIP protease inhibitor (Sigma)). Cells were lysed by Dounce homogenizer and 45 s sonication at 50% cycles and 40% power. Lysate was clarified by centrifugation at

72,000 × g for 45 min using a Beckman Ti45 rotor. The protein was purified by affinity chromatography on HisTrap 5 ml HP column (GE Healthcare) in 50 mM HEPES—pH 7.5, 150 mM NaCl, 10 mM Imidazole, 0.25% CHAPS, 2 mM β-mercaptoethanol and eluted by step imidazole gradient (50–300 mM). Further purification was achieved by size exclusion chromatography on Superose 6 10/300 column (GE Helathcare) in 25 mM HEPES—pH 7.5, 150 mM NaCl, 1 mM DTT.

**Condensate formation assay**. Condensates formation assay was performed as described previously[20,10]. Briefly, mCherry/GFP-p62, mCherry-TAX1BP1 and NBR1 (WT, ΔPB1, ΔUBA, and PB1 domain) were mixed in equimolar amounts (2 μM), unless otherwise specified in the figure legend, in SEC buffer (25 mM HEPES, pH 7.5, 150 mM NaCl, 1 mM DTT). The addition of 5 μM GST-4xUb or ubiquitin chains was used to trigger the condensate formation reaction. After ubiquitin addition time-lapse imaging (1 image every 30 s/1 min for 30 min/1 h) was performed by Spinning disk Microscopy equipped with LD Achroplan 20X/0.4 Corr objective and EM-CCD camera. For the recruitment of FIP200-GFP condensates were formed for 30 min at RT as described above. Then FIP200-GFP (5 μM) was added to the reactions and time-lapse imaging was started.

**Microscopy based protein–protein interaction assay**. Microscopy-based protein–protein interaction assays were performed as described previously[10]. Briefly, Glutathione Sepharose 4B beads (GE Healthcare), with an average diameter of 90 μm, were saturated with GST-tagged bait proteins (4 μg/μl of beads) by incubation at 4 °C for 1 h. Incubation was prolonged to 2 h for GST-FIP200 coupling. Beads were then washed twice in 10x beads volume of washing buffer (25 mM HEPES—pH 7.5, 150 mM NaCl, 1 mM DTT) and resuspend in 1:1 in washing buffer. 2 μM dilutions (unless otherwise stated in the figure legend) of the pray proteins were prepared in the microscopy plate and bait-couple beads were added. After 30 min–1 h incubation at room temperature beads were imaged at the equilibrium at Zeiss LSM 700 confocal microscope equipped with Plan-Apochromat 20X/0.8 WD 0.55 mm objective. For the assay in Fig. 2f RFP-trap beads (Chromotek), with an average diameter of 90 μm, were coupled with mCherry-p62 and the experiment was performed as described above.

**Immunocytochemistry**. For immunocytochemistry analysis, cells were grown on glass cover slips (∅ 12 mm, high precision, Marienfeld-superior) and fixed with 4% (w/v) paraformaldehyde in PBS for 20 min at room temperature. For detection of endogenous LC3B (Fig. 1d), cells were fixed in ice cold methanol for 20 min on ice. Cells were permeabilized in 0.1% Triton X-100 in PBS for 5 min at room temperature, washed twice with PBS and incubated for 1 h at room temperature in blocking buffer (1% BSA in PBS). Subsequently, coverslips were transferred into a humid chamber and incubated with primary antibody (rabbit anti-TAX1BP1 1:100—Cell Signaling, mouse anti-p62 1:100—BD Bioscience, rabbit anti-p62 1:500—MBL, mouse anti-Ubiquitin FK2 1:1000—Enzo Life Science, mouse anti-LC3B 1:100—nanoTools) diluted in blocking buffer for 1 h at room temperature. Following three PBS washing steps, coverslips were incubated, in the dark, with the secondary antibody (goat anti-mouse/rabbit Alexa Fluor 647 1:500—Jackson Immunoresearch, goat anti-mouse/rabbit Alexa Fluor 488 1:1000—Invitrogen) diluted in blocking buffer for 1 h at room temperature. Coverslips were washed three times with PBS and mounted on glass slides (Roth) by inverting them onto a droplet of the mounting media DAPI-Fluoromont-G™ (Southern Biotech).

For p62-FIP200 immunofluorescent labeling in Fig. 6b, cells were permeabilized in 0.25% Triton-X100 for 15 min at room temperature. After two washes in PBS, coverslips were transferred into a humid chamber and incubated with primary antibodies diluted in blocking buffer (1% BSA in PBS) for 1 h at 37 °C (rabbit anti-FIP200 1:200—Cell Signaling, mouse anti-p62 1:100—BD Bioscience). Coverslips were then washed three times for 5 min in PBS and incubated with secondary antibodies (goat anti-mouse Alexa Fluor 488 1:1000—Invitrogen, goat anti-rabbit Alexa Fluor 647 1:500—Jackson Immonoresearch) diluted in blocking buffer for 1 h at 37 °C. After 3 × 5 min washes in PBS, coverslips were mounted on glass slides using DAPI Fluoromont-G™ (Southern Biotech). A complete list of the antibodies used is provided in Supplementary Table 1.

Imaging was performed on a Zeiss LSM 700 confocal microscope equipped with Plan-Apochromat 63x/1.4 Oil DIC objective. To prevent cross-contamination between fluorochromes, each channel was imaged sequentially using the multitrack recording module before merging. Images from fluorescence and confocal acquisitions were processed and analyzed with ImageJ software.

**Live cell imaging**. For live cell imaging, cells were seeded in imaging chambers (Greiner Bio One – 5000 cells/well) and left to grow for 48 h. Cells were then treated as specified in the figure legend and imaged in a temperature- and CO₂-controlled environment with a Visitron Live Spinning Disk microscope (Plan-Apochromat 63x/1.4 Oil DIC objective and an EM-CCD camera). As standard imaging conditions, GFP was imaged with 10% laser power, mScarlet with 14%, LysoTrackerBlue with 3% and iRFP with 10%. Imaging conditions were kept consistent within the same experiment and among replicates. For each condition, 10 different areas of the well were imaged by acquiring 5 μm stacks with a 0.5 μm interval.

**Co-purification of FIP200 with endogenous GFP-NBR1**. HAP1 WT and GFP-AID-NBR1 cells were expanded to 80–90% confluence in 15 cm dishes. Cells were harvested by trypsinization and the cell pellet was washed with PBS and re-suspended in 400 μl of lysis buffer (20 mM HEPES pH 7.5, 250 mM sorbitol, 0.5 mM EGTA, 5 mM Mg-acetate, 0.3 mM DTT, cOmplete EDTA-free protease inhibitor cocktail (Roche)). The cell suspension was flash frozen and left to thaw on ice. The solution was cleared by centrifugation at 1000 × g for 10 min. Protein concentration in the lysate was determined using the Pierce BCA kit (Thermo Scientific). 3500 μg of total protein was brought to 300 μl with wash buffer (25 mM Hepes pH 7.5, 125 mM NaCl, 0.05% Triton X 100) and 3 μl of input were collected. GFP-trap magnetic beads (Chromotek) were washed in wash buffer three times and added to the total protein samples. The samples were incubated for 1 h at 4 °C on a turning wheel. The beads were then separated from the solution using a magnetic rack and 3 μl of unbound fractions were collected. The beads were then washed three times for 5 min with wash buffer. Finally, the beads were resuspended in 10 μl final volume of wash buffer and prepared for SDS-PAGE and western blot analysis.

**Cell treatment with siRNA and drugs**. For siRNA treatment cells were seeded in six-well plates (110,000 cells/well). Cells to be analyzed by immunocytochemistry were seeded on coverslips. Twenty-four hours after seeding cells were transfected with siRNA (20 nM final concentration) using Lipofectamine® RNAiMAX Transfection Reagent (Thermo Fisher) in OptiMEM medium. Forty-eight hours after transfection cells were left untreated or treated as specified in the figure legends and either harvested for western blot analysis or fixed for immuno-fluorescence analysis. When cell treatment with drugs was performed the following conditions were used: bafilomycin (Santa Cruz Biotech. – 400 nM for 2 h); wort-mannin (Sigma – 1 μM for 2 or 3 h, as specified in the figure legend); puromycin (ThermoFisher – 5 μg/ml for 3 h); MG132 (Boston Biochem – 10 μM for 3 h); 1-NAA (Sigma – 500 μM – 1 mM for 3 or 12 h, as specified in the figure legend); doxycycline (50 ng/ml for 12 h).

**Cell lysis and western blotting**. For western blot analysis, cells were harvested with trypsin. Cell pellets were washed with PBS and resuspended in 20 mM Tris-HCl pH 8.0, 10% glycerol, 135 mM NaCl, 0.5% Nonided P-40 Substitute, 2.5 mM MgCl₂, DNase, cOmplete EDTA-free protease inhibitor cocktail (Roche). After 20 min incubation on ice lysates were cleared by centrifugation at 16,000 × g for 5 min at 4 °C and total protein concentration was measured by Bradford protein assay (Bio-Rad). For subsequent western blot analysis 25 μg of lysates were boiled for 5 min at 98 °C and resolved on SDS-PAGE. For the detection of LC3B I and II samples were heated at 60 °C for 10 min. Proteins were then transferred to PVDF membrane by wet blot at 120 V for 90 min. Membranes were blocked in 3% Non-fat dry Milk in TBS + 0.05% Tween-20 (blocking buffer) and incubated O/N at 4 °C with primary antibody diluted in blocking buffer. After 3 × 15 min washes in TBS + 0.05% Tween-20 (TBST), they were incubated with Horse Radish Perox-idase conjugated secondary antibody diluted in blocking buffer for 1 h at RT. After 3 × 15 min washes in TBST, membranes were developed using Super Signal West Pico chemiluminescence substrate. Images were taken with ChemiDoc Touch system (Bio-Rad). Antibodies dilutions used for western blot: rabbit anti-FIP200 (Cell Signaling, 1:1000), mouse anti-NBR1 (Abnova 1:1000, mouse anti-GFP (Roche, 1:1000), mouse anti-mScarlet/RFP (Chromotek, 1:1000), mouse anti-GAPDH (Sigma, 1:25,000), mouse anti-FLAG (Sigma, 1:1000), mouse anti-p62 (BD Bioscience, 1:3000), mouse anti-LC3B (nano-Tools, 1:500), rabbit anti-TAX1BP1 (Cell Signaling, 1:1000). Secondary antibodies: goat anti-rabbit HRP (Jackson Immunoresearch, 1:10,000), goat anti-mouse HRP (Jackson Immunor-esearch, 1:10,000). A complete list of the commercial antibodies used is provided as Supplementary Table 1.

**Quantification and statistical analysis**
*Microscopy based protein–protein interaction assay*. Quantification of microscopy-based protein–protein interaction assays was performed in ImageJ 1.x[36]. A maximum Z-projection of the acquired stack was made and the intensity of the signal around the beads was measured by drawing a line across each bead and taking the maximum gray value along the line. The average values for each sample were averaged between three independent replicates (unless otherwise stated) and plotted with the standard deviations.

**Western blot**. Protein bands intensities were quantified with ImageJ by drawing a rectangle around the gel lane and obtaining the lane profile. The area of the peak in the profile corresponding to the band of interest was taken as a measure of the band intensity. Average intensities (normalized for the intensity of GAPDH or as specified in the figure legends) and standard deviation of three independent experiments were plotted.

**Puncta count and colocalization analysis**. Puncta count and colocalization analysis for immunofluorescence experiments was performed with ImageJ. Images were thresholded and the accuracy of the established threshold was validated manually by comparison with the original image. The same threshold value was applied to all the images of the same experiment. The total number of puncta and

their size was calculated using the "Analyze Particles" function and excluding all particles smaller than 0.05 μm². The number of cells per image was counted manually based on the DAPI staining. The average number of puncta per cell was averaged between three independent experiments and standard errors were calculated.

For colocalization analysis, puncta in both channels were identified with the "Analyze Particles" function as described above and the coordinates saved as regions of interest (ROI). Then the identified puncta from both channels were visualized in different colors on the same image and the overlapping puncta were counted. The number of colocalizing puncta per cell was displayed as the average value from three independent experiments and standard error was calculated.

**Condensate formation assay**. For data extraction from the condensate formation assay a custom ImageJ macro was used as described in[20]. Briefly, images background was subtracted using ImageJ rolling ball algorithm. Then, a manually validated threshold was applied to all images and the number of particles in each field was analyzed using the ImageJ "Analyze Particles" function. Data sets generated with the macro are provided in the Source data file.

**Colocalization analysis for live cell imaging**. For live cell images in Fig. 3 and Supplementary Fig. 3 a macro was designed to quantify the number of particles and total particle area for each channel. The selected ROIs from this macro were then overlapped with the thresholded image from another channel and depending on the end-goal of the analysis, if the ROIs detected signal or did not detect a signal, they were counted further for overlap in the third channel. In this manner, mScarlet particles (representing NBR1) which did not colocalize with lysosomes (LysoTrackerBlue) could be counted as either colocalizing with GFP (p62 particles) or not. Colocalization percentages (normalized as indicated in the figure legends) were shown as average of three independent experiments and standard deviations were calculated.

**Statistical analysis**. For all quantifications an unpaired Student's $t$-test was performed to assess statistical significance. $P$ values for significantly different samples are indicated in the figures.

**Reporting summary**. Further information on research design is available in the Nature Research Reporting Summary linked to this article.

## Data availability
The authors declare that the data supporting the findings of this study are available within the paper and its supplementary information files. Source data are included with this paper. The source data file includes all the datasets for the condensate formation assay and all the uncropped gels and western blots. Source data are provided with this paper.

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

## Acknowledgements
We thank the Max Perutz Labs BioOptics Facility for technical support. This work has been funded by the ERC grant No. 646653, the Austrian Science Fund (FWF P30401-B21 and F79) and a Uni:docs fellowship of the University of Vienna. This project has received funding from the European Union's Framework Programme for Research and Innovation Horizon 2020 (2014-2020) under the Marie Curie Skłodowska Grant Agreement Nr. 847548 to L.F.

## Author contributions
E.T., A.S., F.G., L.F., M.S., and J.R. conducted the experiments. E.T., A.S., and S.M. designed the experiments and wrote the paper.

## Competing interests
S.M. is member of the scientific advisory board of Casma Therapeutics. The other authors declare no competing interests.
