## [Peer Review File · Nature Communications]

REVIEWER COMMENTS

Reviewer #1 (Remarks to the Author):

In the manuscript titled as "Reconstitution defines the roles of p62, NBR1 and TAX1BP1 in ubiquitin condensate formation and autophagy initiation", Turco et al have taken advantage of their in vitro reconstituted ubiquitin condensates system to systematically evaluate the roles of p62, NBR1 and TAX1BP1 in ubiquitin condensate formation and autophagy initiation. Specifically, they have tested wildtype p62, NBR1 or TAX1BP1 and the respective mutants lacking UBA, PB1 or some other concerned domains. Finally, they were able to come up with a sequential recruitment model in which p62, NBR1 or TAX1BP1 drives the other cargo receptors to form and enlarge Ub condensates. The logic and design of whole study was clear and plausible, with all the experiments carried out in most meticulous manner. All these added up to finally address a critical but yet unaddressed question in the field: how the autophagy receptors work concertedly to orchestrate selective autophagic degradation of the substrates.

However, there are several interesting and important questions that remained.

- 1) Among them, in light of the central role of p62-NBR1 heterooligomers in Ub condensate formation, one may wonder what might happen if the PB1 or UBA domain in p62 and NBR1 are respectively swamped with the domain in the other cargo receptor? it is also intriguing to ask the similar question with the other cargo receptors in determining the roles of the respective implicated domains. This reviewer would like to see the authors address these questions with data from at least one actual experiment.
- 2) In further extending the important findings made in this study, could the authors comment on what might be the action modes when the other autophagy/cargo receptors e.g OPTN, NDP52 are involved in the other types of selective autophagy, for example, mitophagy etc.

Taken together, I would suggest a minor revision before final acceptance of the manuscript for publication in Nature Communications.

Reviewer #2 (Remarks to the Author):

The authors investigate the contribution of several autophagy cargo receptors to the formation of ubiquitin condensates and their clearance through selective autophagy.

The authors deploy in vitro reconstitution (in which they are leading experts) as well as cell biological approaches. Notably, the authors generated fluorescently labelled knockin alleles of multiple cargo receptors to avoid overexpression artefacts and enable rapid depletion of proteins. Experiments are performed to high standards and the obtained results support the authors' claims. It must be said, however, that the claims are rather moderate and I am not overtly excited about the actual impact of the manuscript as I fail to see the burning question here. It simply is not clear to me what the authors wanted to discover when they set out on their journey. The most interesting discovery is the apparent cooperativity between different cargo receptors in the condensation and removal of ubiquitinated proteins.

Specific points:

1. The authors refer to NBR1 as carrying a 'high affinity UBA' domain. The statement should be refined by stating the ligand (and by making clear that only a limited number of ub linkage types have been measured and even those not too particularly high standards, implying that a 'low affinity UBA domains' (as in p62) may well have unknown higher affinity ligands).

2. Figure 1: Aggregates are labelled as “ub condensates”. While the dots contain ub, there is no obvious reason that their assembly is necessarily ub-driven.

Reviewer #3 (Remarks to the Author):

In this manuscript, authors use in vitro reconstitution and cell biology to define the hierarchy role of P62, NBR1 and TAX1BP1 in p62 droplet formation and recruitment of autophagy machinery in selective autophagy of ubiquitinated proteins. Overall, it is an elegant study which definitively define the role of these proteins in selective autophagy, this study is well designed and data are compelling, this reviewer has only a few suggestions to further improve this manuscript.

- 1) Given the fact TAX1BP1 can inhibit p62 droplet formation in vitro, does knockdown TAX1BP1 enhance p62 droplet formation?
- 2) Figure 7, the model is a little bit too complicated, a simpler model focus on the role of P62, NBR1 and TAX1BP1 on selective autophagy of ubiquitinated proteins maybe more effective.
- 3) Figure 2B, the size of scale bar should indicated in figure legend, and reader will benefit from larger magnification of image, the shape of droplet can't be appreciated under the current magnification.
- 4) Figure 6B, the signal from FIP200 is very weaker, authors should try to improve the signal of FIP200.

Response to reviewers

Reviewer #1 (Remarks to the Author):

In the manuscript titled as “Reconstitution defines the roles of p62, NBR1 and TAX1BP1 in ubiquitin condensate formation and autophagy initiation”, Turco et al have taken advantage of their in vitro reconstituted ubiquitin condensates system to systematically evaluate the roles of p62, NBR1 and TAX1BP1 in ubiquitin condensate formation and autophagy initiation. Specifically, they have tested wildtype p62, NBR1 or TAX1BP1 and the respective mutants lacking UBA, PB1 or some other concerned domains. Finally, they were able to come up with a sequential recruitment model in which p62, NBR1 or TAX1BP1 drives the other cargo receptors to form and enlarge Ub condensates. The logic and design of whole study was clear and plausible, with all the experiments carried out in most meticulous manner. All these added up to finally address a critical but yet unaddressed question in the field: how the autophagy receptors work concertedly to orchestrate selective autophagic degradation of the substrates.

We thank the reviewer for the positive assessment of our work.

However, there are several interesting and important questions that remained.

1) Among them, in light of the central role of p62-NBR1 heterooligomers in Ub condensate formation, one may wonder what might happen if the PB1 or UBA domain in p62 and NBR1 are respectively swamped with the domain in the other cargo receptor? it is also intriguing to ask the similar question with the other cargo receptors in determining the roles of the respective implicated domains. This reviewer would like to see the authors address these questions with data from at least one actual experiment.

We thank the reviewer for this suggestion. Following the reviewer’s advice, we replaced the UBA domain of p62 with the NBR1 UBA domain and tested this protein in our condensate formation assay. In the experiment shown in Supplementary fig. 2I, the chimeric p62 protein containing the NBR1 UBA domain produced a higher number and larger condensates compared to wild type p62. The addition of NBR1 to wild type p62 enhanced condensate formation as shown before. In contrast, condensate formation by the chimeric p62 with the high affinity NBR1 UBA was not further stimulated by the addition of NBR1.

2) In further extending the important findings made in this study, could the authors comment on what might be the action modes when the other autophagy/cargo receptors e.g OPTN, NDP52 are involved in the other types of selective autophagy, for example, mitophagy etc.

In a previous study (Zaffagnini, 2018, EMBOJ) we have shown that Optineurin is not recruited to p62 – ubiquitin condensates. So far, we have not tested NDP52. However, it is possible that they play roles in condensate turnover in cells. One of the main differences of Optineurin and NDP52 when compared to p62 and NBR1 is that their roles in mitophagy and xenophagy do not require the accumulation of the cargo material in condensates as the cargo is already a bulky structure. We discuss this briefly in the revised manuscript (page 13).

Taken together, I would suggest a minor revision before final acceptance of the manuscript for publication in Nature Communications.

Reviewer #2 (Remarks to the Author):

The authors investigate the contribution of several autophagy cargo receptors to the formation of ubiquitin condensates and their clearance through selective autophagy.

The authors deploy in vitro reconstitution (in which they are leading experts) as well as cell biological approaches. Notably, the authors generated fluorescently labelled knockin alleles of multiple cargo receptors to avoid overexpression artefacts and enable rapid depletion of proteins. Experiments are performed to high standards and the obtained results support the authors' claims. It must be said, however, that the claims are rather moderate and I am not overtly excited about the actual impact of the manuscript as I fail to see the burning question here. It simply is not clear to me what the authors wanted to discover when they set out on their journey. The most interesting discovery is the apparent cooperativity between different cargo receptors in the condensation and removal of ubiquitinated proteins.

We thank the reviewer for the overall positive comments on our manuscript.

Specific points:

1. The authors refer to NBR1 as carrying a 'high affinity UBA' domain. The statement should be refined by stating the ligand (and by making clear that only a limited number of ub linkage types have been measured and even those not too particularly high standards, implying that a 'low affinity UBA domains' (as in p62) may well have unknown higher affinity ligands).

We thank the reviewer for pointing this out. In Walinda et al., 2014 the affinity of NBR1 and p62 UBA domains for monoubiquitin as well as K48- and K63-linked di-ubiquitin was compared. We have now changed the text to make clear that monoubiquitin is the ligand to which the "high affinity" attribute refers to (page 6).

2. Figure 1: Aggregates are labelled as "ub condensates". While the dots contain ub, there is no obvious reason that their assembly is necessarily ub-driven. Thank you for this observation. To avoid misunderstanding we have now replaced "ub condensates" with "condensates" or with "ubiquitin containing condensates" in the text referring to Figure 1.

Reviewer #3 (Remarks to the Author):

In this manuscript, authors use *in vitro* reconstitution and cell biology to define the hierarchy role of P62, NBR1 and TAX1BP1 in p62 droplet formation and recruitment of autophagy machinery in selective autophagy of ubiquitinated proteins. Overall, it is an elegant study which definitively define the role of these proteins in selective autophagy, this study is well designed and data are compelling, this reviewer has only a few suggestions to further improve this manuscript.

We thank the reviewer for the positive assessment of our work.

1) Given the fact TAX1BP1 can inhibit p62 droplet formation *in vitro*, does knockdown TAX1BP1 enhance p62 droplet formation?

The reviewer raises an interesting point. Indeed, *in vitro*, the number of p62 condensates in presence of TAX1BP1 is reduced. In cells we observe an increased number of p62 puncta upon TAX1BP1 knockdown (we quantified the number of p62 puncta/cell in HAP1 cells treated with TAX1BP1 siRNA in the context of the experiment shown in figure 6). Of course, the effect of TAX1BP1 depletion on the number and size of p62 condensates is complicated by the fact that their turnover is also reduced.

2) Figure 7, the model is a little bit too complicated, a simpler model focus on the role of P62, NBR1 and TAX1BP1 on selective autophagy of ubiquitinated proteins maybe more effective. Thank you for the suggestion. We have simplified the model by removing the upper part.

3) Figure 2B, the size of scale bar should indicated in figure legend, and reader will benefit from larger magnification of image, the shape of droplet can't be appreciated under the current magnification.

Thank you for pointing this out. We have added the information related to the scale bar in the figure legend and we provide now a larger magnification of the images. We would like to point out that the condensates formed by p62 and ubiquitin are peculiar structures as the p62 oligomers/filaments are rather static in them, while the ubiquitin is highly mobile (Zaffagnini et al. EMBO J., 2018). Therefore, they tend to have more irregular shapes compared to condensates/droplets formed by canonical liquid-liquid phase separation reactions.

4) Figure 6B, the signal from FIP200 is very weaker, authors should try to improve the signal of FIP200.

The signal in Fig. 6B was improved and FIP200 puncta can be better appreciated in the revised figure.

REVIEWERS' COMMENTS

Reviewer #3 (Remarks to the Author):

Authors addressed all of my queries satisfactorily. Congratulations for this elegant work.

REVIEWERS' COMMENTS

Reviewer #3 (Remarks to the Author):

Authors addressed all of my queries satisfactorily. Congratulations for this elegant work.

We thank the reviewer for the positive assessment of our work.